# A NEW PERSPECTIVE IN UNDERSTANDING OF ADAM-TYPE ALGORITHMS AND BEYOND

## ABSTRACT

First-order adaptive optimization algorithms such as Adam play an important role in modern deep learning due to their super fast convergence speed in solving large scale optimization problems. However, Adam's non-convergence behavior and disturbing generalization ability make it fall into a love-hate relationship to the deep learning community. Previous studies on Adam and its variants (refer as Adam-Type algorithms) mainly rely on theoretical regret bound analysis, which overlooks the natural characteristics residing in Adam-Type algorithms and limits our understanding. In this paper, we aim at seeking a different interpretation of Adam-Type algorithms so that we can intuitively comprehend and improve them. The way we chose is based on a traditional online convex optimization algorithm scheme known as mirror descent method. By bridging Adam and mirror descent algorithm, we receive a clear map of the functionality of each term in Adam. In addition, this new angle brings us a new insight on identifying the non-convergence issue of Adam and explaining the superior convergence speed of Adam than other first-order methods. Moreover, we provide a new variant of Adam-Type algorithm, namely AdamAL which can naturally mitigate the non-convergence issue of Adam and improve its performance. We further conduct experiments on various popular deep learning tasks and models, and the results are quite promising.

## 1 INTRODUCTION

In recent years, first-order optimization algorithms with adaptive learning rate have become the dominant method to train deep neural networks because these methods show extraordinary power in solving large-scale machine learning optimization problems. By cooperating with first-order information, adaptive methods iteratively update parameters by moving them to the direction of the negative gradient of the cost function with non-fixed learning rate. The first algorithm in this line of research can be dated back to McMahan & Streeter (2010), where they adaptively choose regularization functions for bounding objective function parameters based on the loss functions observed at each iteration. Then, in Streeter & McMahan (2010), they demonstrate that the convergence rates can often be dramatically improved through the use of preconditioning. The insight of these two methods is parallel but effective, that is, they try to modify the gradients' magnitude with adaptive pre-coordinate learning rates. Later, AdaGrad (Duchi et al., 2011) carefully chooses the preconditioner and provides the first practical adaptive algorithm with a tight theoretical guarantee based on Zinkevich (2003) regret analysis. Although AdaGrad achieves a great success in the sparse settings, the rapid decay of the adaptive learning rate limits its usage. This is due to AdaGrad accumulating all the past gradients as its learning rate. And this unbounded learning rate becomes extremely small when one has a large number of training iterations. To address this, several variants of AdaGrad, such as AdaDelta (Zeiler, 2012) and RMSProp (Hinton et al., 2012) have been proposed to mitigate the rapid decay of the learning rate. Based on these, Adma (Kingma & Ba, 2014) incorporating with first momentum correction accelerates the convergence speed of first-order optimization algorithms to a new height. The use of exponential moving average (EMA) in Adam becomes a key to Adam's success. Up to this point, we can summarize the adaptive or non-adaptive first-order optimization algorithms as follows.

Denote $g_t \in \mathcal{R}^d$ as the gradient of generic optimization problem $f$ with respect to its parameters $x \in \mathcal{R}^d$ at iteration $t$, then the generic updating rule of adaptive methods can be expressed as (Reddi

et al., 2019):

$$x_{t+1} = x_t - \frac{\eta_t}{\sqrt{v_t}} \odot m_t \tag{1}$$

where $\odot$ denotes the entry-wise or Hadamard product and the $\alpha_t$ is the base learning rate. In the equation above, $m_t = \vartheta(g_1, \cdots, g_t)$ is a function related to the historical gradients up to $t$; and $v_t = \upsilon(g_1, \cdots, g_t)$ is a function that produces a d-dimensional vector with non-negative entry; The design of $v_t$ is simple of non-adaptive methods, such as vanilla stochastic gradient descent (Vanilla SGD), where we have $v_t = \boldsymbol{I}$, however, it is crucial for Adam-Type algorithms. For Adam, in particular, the $m_t$ and $v_t$ are computed by EMA of gradient, with coefficient $\beta_1$ and $\beta_2$ where

$$m_t = (1 - \beta_1) \sum_{i=1}^{i=t} \beta_1^{t-i} g_i \quad \text{and} \quad v_t = (1 - \beta_2) \sum_{i=1}^{i=t} \beta_2^{t-i} g_i^2 \tag{2}$$

Adam, the most popular adaptive method, has been widely adopted by the deep learning community. The root cause of the fast convergence of Adam and its variants in convex or non-convex optimization problems remains an open question (Chen et al., 2018). In addition, the generalization ability and out-of-sample behavior of Adam-Type methods are even worse than traditional non-adaptive counterparts such as Vanilla SGD (Wilson et al., 2017).

In order to understand the insight behind those adaptive algorithms and close the generalization gap, several Adam-Type algorithms have been proposed including (Reddi et al., 2019; Luo et al., 2019; Zhou et al., 2018b; Balles & Hennig, 2017; Liu et al., 2019). Although they propose many different kinds of viewpoints in understanding the performance of Adam and demonstrate a series of correction methods to improve Adam, we think the mechanism of Adam-Type algorithms is still unclear. For example, one common thinking about about $m_t$ and $v_t$ in Adam is first and second moments of unbiased estimator $g_t$, however, why this second moment can be used as adaptive learning rate? Does the $v_t$ really behave as the second moment of the gradient $g_t$? Also, another commonly asked question is where the Adam adopts such a faster convergence speed than other first-order methods?

These mysterious questions on adaptive methods finally leads us to rethink the Adam-Type algorithms in a neutral way. In this paper, we provide a new insight into Adam-Type algorithms, which brings a new perspective of comprehension on Adam-Type algorithms and it allows us to easily identify the misbehavior of Adam such as non-convergence issues. In the aforementioned works, the analysis of Adam-Type methods is based on Kingma & Ba (2014) framework, which limits our understanding. In fact, when we look back to the origin of adaptive learning rate methods mentioned in Streeter & McMahan (2010), we notice that the design of Adam is highly related to adaptive regularization of follow-the-proximally-regularized-leader (FTPRL) method and it can also be regarded as a variant of traditional mirror descent method (Xiao, 2010). The more detail and our motivation can be found in the next section. For simplicity, we use standard Adam as our touchstone through the paper to convey our main idea, and their variants follow the same thoughts. Now, we summarize our contribution in three folds:

1. We provide a new perspective in understanding the non-convergence behavior of Adam-Type algorithms based on mirror descent approach. Our analysis agrees well with the previous works but much more intuitive and effective.

2. Based on our observation, we identify potential faults in Adam-Type algorithms and we provide a new Adam variant algorithm, named AdamAL.

3. We conduct a series of experiments on different machine learning tasks and models by using our AdamAL algorithm, and the results are promising and the performance of AdamAL is never worse than Adam.

## 2 PRELIMINARIES AND MOTIVATIONS

**Notations** Given a vector $x \in \mathcal{R}^d$ we denote its i-th entry by $x_i$; We use $||x||$ to denote its $l_2$ norm; for a vector $x_t$ in the t-th iteration, the i-th coordinate of $x_t$ is denoted as $x_{t,i}$. We also define $x_{1:t} = \sum_{i=1}^{t} x_i$. Given two vectors $x, y \in \mathcal{R}$, we use $\langle x, y \rangle$ to denote their inner product, $x \odot y$ to denote element-wise product, $\frac{x}{y}$ to denote entry-wise division, the $\max(x, y)$ to denote entry-wise maximum and $\min(x, y)$ to denote entry-wise minimum. We use $\mathcal{S}_+$ to denote the set of all positive definite matrices $M$. We use $M^{\frac{1}{2}}$ to denote $\sqrt{M}$.

## 2.1 PRELIMINARIES

**Nonlinear projected subgradient methods and mirror descent algorithm** Iterative gradient descent (GD) scheme, which can be traced back to (Cauchy, 1847), is the simplest strategy to minimize convex optimization problems. It was further developed as Zinkevich Online Greedy Subgradient Project (OGSP) (Zinkevich, 2003), which can be considered as a variation of project gradient descent (PGD) algorithm with following updating rules:

$$x_{t+1} = P_{\mathcal{X}}(x_t - \eta f^{'}(x_t)) \text{ (standard PGD)} \quad x_{t+1} = P^A_{\mathcal{X}}(x_t - \eta_t g_t) \text{ (Zinkevich OGSP)} \quad (3)$$

where $P^A_{\mathcal{X}}$ is a distance-based projector denoting the projection of a point $y$ onto $\mathcal{X}$ by $P^A_{\mathcal{X}}(y) = \arg\min_{x \in \mathcal{X}} ||x - y||_A$, and $|| \cdot ||_A = \langle x, Ax \rangle$, and $g_t \in \partial f_t(x_t) = f^{'}(x_t)$ is the subgradient of the objective function $f_t(\cdot)$. The well-known bottleneck of using PGD is that it can only work in Hilbert space $\mathcal{H}$ and it cannot be extended to more general cases (in modern machine learning) of optimization plays in some Banach space $\mathcal{B}$ where the Euclidean norms cannot be computed. To this end, the mirror descent algorithm (MDA) introduced by (Nemirovsky & Yudin, 1983) overcomes such infeasibility by using the linearity on dual vector space $\mathcal{B}^*$ and a carefully designed mirror map (Bubeck, 2014). Hence, MDA has following updating schemes:

$$x_{t+1} = \arg\min_{x \in \mathcal{X}}\{\langle f^{'}(x_t), x \rangle + \frac{1}{\eta_t} D_\psi(\cdot)\} \quad (4)$$

the nonlinear projection therefore encoded in the term $D_\psi$. By replacing the Euclidean quadratic norms in Equation. 3 with more general distance-liked settings such as Bregman distance function $D_\psi(\cdot)$ define on $\psi(\cdot)$, the equivalence of PGD algorithm and MDA has been proved by Beck & Teboulle (2003). For example, the simplest version of MDA is that: taking $\psi(x) = \frac{1}{2}||x||^2_2$, and $D_\psi(x, x_t) = \frac{1}{2}||x - x_t||^2_2$ then plugging into Equation. 4, we have:

$$x_{t+1} = \arg\min_{x \in \mathcal{X}}\{\langle g_t, x \rangle + \frac{1}{2\eta_t}||x - x_t||^2_2\} \quad (5)$$

Update scheme in the above equation. 5 is also known as **proximal point algorithm** (Rockafellar, 1976). The equivalence of Equation. 5 and PGD can be achieved by taking the derivative of our target function on $x$, and rearranging the formula:

$$x_{t+1} = x^* = x_t - \eta_t g_t = P^I_{\mathcal{X}}(x_t - \eta_t g_t) \quad (6)$$

Now, we see the last two expressions in Equation. 6 are well-known subgradient descent update. We restate the proposition 3.2 in Beck & Teboulle (2003) as:

**Proposition 1.** *Assume $\mathcal{X}$ is a closed convex subset in $\mathcal{R}$ with non-empty interior, and objective function $f : \mathcal{X} \to \mathcal{R}$ is a convex and Lipschitz function. Suppose the optimal set of $x$ denoted by $\mathcal{X}^*$ is non-empty, we can compute the subgradient of $f$ at $x$ as $g \in \partial f(x)$. For a convex mirror mapping function $\psi : \mathcal{X} \to \mathcal{R}$ with conjugate function $\psi^*$ defined by $\psi^*(y) = \max_{x \in \mathcal{X}}\{\langle x, y \rangle - \psi(x)\}$. Then the sequence $\{x_t\} \subseteq \mathcal{X}$ generated by MDA is equivalent to the sequence generated by PGD.*

We state the equivalence of PGD algorithm and MDA, particularly, the general gradient descent (or SGD) can be directly derived from the Equation. 6.

**Follow the proximally-reqularized leader (FTPRL)** FTPRL is introduced by McMahan & Streeter (2010) belongs to the family of follow-the-regularized-leader (FTRL) algorithms such as Regularized Dual Averaging (Xiao, 2010). In general, FTRL-Type has following update rule:

$$x_{t+1} = \arg\min_{x \in \mathcal{X}}\{(\sum_{\tau=1}^{t} f^{'}_\tau(x_\tau)) \cdot x + R_{1:t}(x) \quad (7)$$

where the subgradient of objective function $f^{'}_\tau(x_\tau)$ is approximated by the gradient at $x_\tau$ and $R_{1:t}(x)$ is defined as regularization. Particularly, the formal FTPRL is:

$$x_{t+1} = \arg\min_{x \in \mathcal{X}}\{g_{1:t} \cdot x + \phi_{1:t} \cdot x + \Psi(x) + \frac{1}{2}\sum_{\tau=1}^{t}||Q^{\frac{1}{2}}_\tau(x - x_\tau)||^2_2\} \quad (8)$$

with $\phi_{1:t} \cdot x + \Psi(x)$ can be considered as non-smooth composite term which is orthogonal to our

Table 1: An overview of first-order optimization methods using the generic framework

|  | SGD | SGDM | AdaGrad | RMSProp | Adam |
|---|---|---|---|---|---|
| $m_t$ | $g_t$ | $\sum_{i=1}^{t} \beta^{t-i} g_i$ | $g_t$ | $g_t$ | $(1-\beta_1)\sum_{i=1}^{t} \beta_1^{t-i} g_i$ |
| $v_t$ | $\boldsymbol{I}$ | $\boldsymbol{I}$ | $\sum_{i=1}^{t} g_i^2$ | $(1-\beta)\sum_{i=1}^{t} \beta^{t-i} g_i^2$ | $(1-\beta_2)\sum_{i=1}^{t} \beta_2^{t-i} g_i^2$ |
| lr | $\eta_t$ | $\eta_t$ | $\eta_0 \frac{1}{\sqrt{v_t}}$ | $\eta_0 \frac{1}{\sqrt{v_t}}$ | $\eta_0 \frac{1}{\sqrt{v_t}}$ |

paper, more details can be found in McMahan (2010b). The last term in the above equation is stabilizing regularization to ensure low regret. We mention that the $Q_\tau$ can be either regarded as **scale of regularization** or **generalized learning rate** which plays crucial role in our paper. As we can see, FTPRL appears quite different from MDA stated in Equation. 5, however, in McMahan (2010a;b) they show that in the case of selecting quadratic stabilizing regularization, the FTPRL and generalized MDA only has differences in parameter centering. In fact, MDA is illustrated in the Equation. 4 regularizing the parameter to be close to the origin, on contrast, FTPRL is regularizing the parameter at current feasible point. No surprising, McMahan (2010a) propose the equivalence proof of FTPRL and a variation algorithm of the MDA as follow.

**Proposition 2.** *(McMahan, 2010a) Let $R_t$ be a sequence of differentiable convex functions ($\nabla R_t(0) = 0$), and let $\Psi$ be an arbitrary convex function. Define the proximal-MDA with updating rule:*

$$x_{t+1} = \underset{x \in \mathcal{X}}{\arg\min}\{\langle g_t(x_t), x \rangle + \Psi(x) + D_{R_{1:t}}(\cdot)\} \tag{9}$$

*where the Bregman distance function $D_{R_t}$ with respect to $R_t$ where $R_t(x) = R_t(x - x_t)$. And applying FTPRL to the same objective function, with:*

$$x_{t+1} = \underset{x \in \mathcal{X}}{\arg\min}\{g_{1:t} \cdot x + \phi_{1:t} \cdot x + \Psi(x) + R_{1:t}\} \tag{10}$$

*when $\phi_t \in \partial\Psi$, such that $g_{1:t} + \phi_{1:t} + \nabla R_{1:t}(x_t) = 0$. Then the two above update scheme are equivalent.*

At this moment, we state a series of equivalence proposition from FTPRL to proximal-MDA (variation of MDA) and MDA to PGD. Now, we can construct the equivalence proof of FTPRL and PGD properly. Although the direct proof of equivalence between FTPRL and PGD is provided unofficially in McMahan (2010a), we would like to elaborate the intuition behind each algorithm and deduce our perspective in understanding the Adam-Type algorithms.

## 2.2 MOTIVATION

**Non-Adam-type and Adam-type algorithms to mirror descent (FTPRL)** Non-Adam-type online gradient descent algorithms including SGD, SGD with momentum, Polyak's HB and Nesterov's accelerated gradient method can be easily understood as first order optimization with different momentum function. And, interestingly, most of them have physical interpretations. However, as shown in Table 1, Adam-Type algorithms do not rely on the non-increasing learning rate when iteratively updating their parameters. Instead, they perform adaptively learning rate element-wise on parameters according to the first order information. However, this leads to the most mysterious part in the Adam-Type algorithm where the adaptive update is represented as $-\frac{\eta_0}{\sqrt{v_t}} \odot m_t$. One common interpretation of this updating representation is regarding $-\frac{\eta_0}{\sqrt{v_t}}$ as adaptive learning rate, and regarding $m_t$ as general first order gradient. However, we **can not** treat Adam-Type algorithm in this way because the first order information $g_t$ resides in both $m_t$ and $v_t$ and we are unable to simply decorrelate them as common gradient with step-size scheme. Another possible interpretation of such updating is related to the Newton's second order method, but there is no free lunch for expressing in this way. We will discuss it later.

In order to dissect Adam-Type algorithm, we recall the FTPRL algorithm mentioned in the previous section. A natural question arises: can we interpret Adam-Type algorithm as a variant of mirror descent method? Before answering the question, let us explain why we present Adam-Type algorithm as MDA is beneficial. We summary below:

Table 2: An overview of first-order optimization methods using Mirror Descent expression

| | SGD | SGDM | AdaGrad | Adam |
|---|---|---|---|---|
| $A$ | $g_{1:t} \cdot x$ | $(g_{1:t} - \sum_{i=1}^{i=t,j=t} \beta^{j+1-i}) \cdot x$ | $g_t \cdot x$ | $(g_{1:t} - \sum_{i=1}^{i=t,j=t} \beta^{j+1-i}) \cdot x$ |
| $C$ | $\frac{1}{2}\|x - x_\tau\|_2^2$ | $\frac{1}{2}\|x - x_\tau\|_2^2$ | $\frac{1}{2}\sum_{\tau=1}^{t}\sigma_\tau\|x - x_\tau\|_2^2$ | $\frac{1}{2}\sum_{\tau=1}^{t}\sigma_\tau\|x - x_\tau\|_2^2$ |
| $\rho$ | $1$ | $\frac{1}{1-\beta}$ | $1$ | $1$ |

* Term $A$, $C$, $\rho$ are defined in Equation 11. See below.

1. **Implicit updates:** This concept was first derived by Kivinen & Warmuth (1997) and later pointed by Kulis & Bartlett (2010). It refers to without using explicit first-order update rules to efficiently compute the parameters. The mirror descent algorithms, in fact, adopt implicit update rules very well, therefore, the explicit update will roll into some regularized-liked terms which can elaborate more insights. In contrast, like Adam, the learning rate is explicitly defined as $\frac{\eta_0}{\sqrt{v_t}}$ which is hard to identify from intuition unless relying on the theoretical proofs.

2. **First-order information dissection** The entire adaptive update involves in Hadamard product and division. Both numerator and denominator are highly related to the construction function with respect to the first-order gradient $g$. The underlying relation between two functions $m_t$ and $v_t$ can not be decorrelated. However, as shown in Table 2, we represent Adam-Type algorithms in MDA way so that we can separate the numerator and denominator as simple additive scheme where the hard-understanding division disappears and meanwhile, the alignment of $m_t$ and $v_t$ gone.

3. **Equivalence guarantee** The original adaptive method AdaGrad is build upon non-linear subgradient projection, and Adam actually inherit such design indicate by their regret proof. To re-represent Adam-Type algorithm as FTPRL style, we do require the theoretical equivalence guarantee so that we can transform safely. As we discuss in previous (Proposition 1& 2), we successfully build such a bridge in between two types of algorithm.

In general, these first-order algorithms can be written in FTPRL style (also see Table 2):

$$x_{t+1} = \underset{x \in \mathcal{X}}{\arg\min}\{\rho \underbrace{(g_{1:t} + C_{1:t}) \cdot x}_{A} + \underbrace{\Psi(x)}_{B} + \underbrace{\frac{1}{2}\sum_{\tau=1}^{t}\|Q^{\frac{1}{2}}(x - x_\tau)\|_2^2}_{C}\} \qquad (11)$$

The understanding of above representation is highly related to our analysis on Adam-Type algorithms. Term A has two parts, the first part $g_{1:t} \cdot x$ is an approximation to $f_{1:t}$ based on the gradient; the second part $C_{1:t} \cdot x$ refers to *first-order momentum correction* or *fault tolerant* in this literature. Term B is similar to the setting of FTPRL, but we usually consider it as $l_2$-regularization. In addition, term C stabilizing regularization plays a crucial role in this transformation because (1) the implicit updates from $x_t$ to $x_{t+1}$ happen in this term; (2) the $Q_\tau$ residing in norms can be regarded as generalized learning rate or even more complicated format; (3) the rule of parameter centering to the current feasible solution is figured. Finally, the leading $\rho$ is a *balancing coefficient*, aims at controlling the tendency of minimization between term A and C. Smaller $\rho$ value will guide the minimization process and relies more on term C, otherwise term A will dominate the minimization.

The simplest format transformation from Vanilla SGD to MD is illustrated in Equation. 5. Now, we rewrite it as FTPRL style according to Equation. 8 (for the sake of simplicity, we do not consider the $\Psi$):

$$x_{t+1} = \underset{x \in \mathcal{X}}{\arg\min}\{g_{1:t} \cdot x + \frac{1}{2}\sum_{\tau=1}^{t}\sigma_\tau\|x - x_\tau\|_2^2\} \qquad (12)$$

where the $\rho = 1$, $C_{1:t}$ is zero, and $Q_\tau$ sets to be $Q_\tau = \sigma_\tau^2 I$. If Vanilla SGD with learning rate $\eta_t = \frac{1}{t}$, the $\sum_{\tau=1}^{t}\sigma_\tau = t$. If it with fixed learning rate $\eta_t = \frac{1}{k}$, then $\sum_{\tau=1}^{t}\sigma_\tau = k$. To be

mentioned here, fixed learning rate SGD is kind of special, it can be easily explained as Equation. 5, however, in FTPRL format (Equation. 12), we need to be more careful.

Mirror descent like Vanilla SGD with non-increasing learning rates shows the great insight of understanding current algorithms. First, we are always looking for the next step $x_{t+1}$ in the opposite direction of the current gradient because $\cos(\pi) = -1$ minimize Equation. 12. Second, $x_{t+1}$ is being bound in a region centered at previous step $x_t$. This is due to the fact that we do not want the new solution to be far away from the current feasible solution, otherwise, may cause slow convergence, and McMahan (2010b) confirm this view. Third, $\sum_{\tau=1}^{t} \sigma_\tau$ should be non-decreasing alone time. Similar, when $\sigma_\tau$ gets smaller, the bounding region expands, reducing the convergence speed.

## 3 NEW PERSPECTIVE ON ADAM

In this section, we deliver a new perspective on Adam-Type algorithm from the Mirror Descent point of view. In this lecture, we mainly focus on Adam to demonstrate our analysis, for the other variants, they have similar results.

### 3.1 ADAM ON MIRROR DESCENT

According to the Table 2, the subgradient projected Adam can be written as follows (entry-wise). In order to make a clear comparison, we also show SGDM alongside.

$$\text{Adam: } x_{t+1,i} = \arg\min_{x \in \mathcal{X}} \{ (g_{1:t,i} - \sum_{\tau=1}^{\tau=t,j=t} \beta_1^{j+1-i} g_{\tau,i}) \cdot x_{,i} + \frac{1}{2} \sum_{\tau=1}^{t} \sigma_{\tau,i} ||x_{,i} - x_{\tau,i}||_2^2 \}$$

$$\text{SGDM: } x_{t+1,i} = \arg\min_{x \in \mathcal{X}} \{ \frac{1}{1-\beta_1} (g_{1:t,i} - \sum_{\tau=1}^{\tau=t,j=t} \beta_1^{j+1-i} g_{\tau,i}) \cdot x_{,i} + \frac{1}{2} \sum_{\tau=1}^{t} \sigma_\tau^* ||x_{,i} - x_{\tau,i}||_2^2 \}$$

(13)

where both $\beta_1 \leq 1$ (commonly chose $\beta_1 = 0.9$), the $\sum_{\tau=1}^{t} \sigma_{\tau,i} = \sqrt{(1-\beta_2)\sum_{\tau=1}^{\tau=t} \beta_2^{t-\tau} g_{\tau,i}^2}$ in Adam settings with $\beta_2 = 0.999$. In SGDM, $\sum_{\tau=1}^{t} \sigma_\tau^*$ performs differently, (1) SGDM is non-adaptive method, the $\tau^*$ applies to all entry on the $x$; (2) $\sum_{\tau=1}^{t} \sigma_\sigma^* = t$ if we have learning rate $\eta_t = \frac{1}{t}$, it also means $\sigma^* = 1$ for all $\tau \in T$. First, let us recall Vanilla SGD and compare it with SGDM:

**Corollary 1.** *Compare to Vanilla SGD, SGDM employs first-order momentum correction defined in Section. 2.2 to correct the possible wrong direction prediction, making a smooth optimization trajectory. (This is a well known truth. We verify it by experiments show in Appendix.)*

Besides, we also notice that Adam has $\rho = 1$ while SGDM has $\rho = \frac{1}{1-\beta_1} = 10$ (a common setting of $\beta = 0.9$). Recall the previous section, by definition of $\rho$, we have following corollary:

**Corollary 2.** *Compare to Vanilla SGD or Adam, SGDM has larger $\rho$ value indicates that the SGDM is **more sensitive on the value change of loss function**.*

To explain this, we know one of the drawbacks of SGDM is that SGDM is prone to oscillation around the optimal point, because SGDM has relatively weak bound in proximal terms, and is very sensitive to small change of loss, that is, $\rho$ factor amplifies this loss change up to tenfold. We notice that Adam avoids such imbalance between objective function and its parameter regularization by using EMA wisely.

Now, back to Adam, before we move forward, we define some terms in the Equation. 13.Adam,

> **Proximal Searching Region** refers to $D = ||x - x_\tau||_2^2$, this Euclidean quadratic norm reflects the geometry of given constraints feasible set $\mathcal{X}$. We can also treat it as a regularization process. This region is inversely proportional ($D \propto \frac{1}{B}$) to the Regularization Budget defined below.

> **Regularization Budget** refer to $B = \sum_{\tau=1}^{t} \sigma_{\tau,i}$, it indicates total "weight" we can distribute to the Proximal Searching Region. More weight it has been given will lead to a stronger regularization in bounding, and a small search space centering at $x_\tau$.

In Hoffer et al. (2017), they define an ultra slow diffusion phenomenon when they evaluate the distance from current weight to initial weight point with $||x_t - x_0||_2^2 \sim \log t$. Interestingly, this result is entirely consistent with our Proximal Searching Region analysis, because for any $t \in T$, we have $||x_{t+1} - x_t|| \sim (\log t)' = \frac{1}{t}$. Now, we summarise our results as follows:

1. **Hyper-parameter $\beta_1$:** $\beta_1$ exponential smooth **eliminates** the presence of **imbalance** in between the goal of minimizing loss of function and the constraints of searching in proximal region. In other words, Adam treats both conditions fairly.

2. **First-order momentum:** the usage of $m_t$ leads to a smooth optimization trajectory which avoids the sharp twist such as SGD. It can benefit Adam if searching in the wrong direction, the momentum correction $C_\tau$ can compensate the party of penalty directly from the loss function.

3. **Adam-Type algorithm:** Adam-Type algorithms such as AdamAL (this paper), AMSGrad AdaBound, AdaShift, NosAda, etc. can be regarded as making a correction on one of regularization term (most on $||Q_\tau^{\frac{1}{2}}|| ||x - x_\tau||_2^2$).

4. **Learning rate:** mirror descent corporate with implicit update makes learning rate (step size) act as a scalar factor of regularization.

By transferring the Adam to MDA-liked method, we successfully disassemble the Adam updating $-\frac{m_t}{\sqrt{v_t}}$ into two additive terms and identify their functionality separately. We mainly focus on the $m_t$ part in this section, we will move our eye on $\frac{1}{\sqrt{v_t}}$ in the next section.

## 3.2 ADAM $V_t$ AND THE NON-CONVERGENCE OF ADAM

A common thinking of Adam's adaptive learning rate $\frac{1}{\sqrt{v_t}}$ will treat $v_t$ as second moment approximation. However, in our perspective, we regard it as **adaptive regularize scalar** and it performs implicit updates by replacing the explicit learning rate. Two adaptive regularize from AdaGrad and Adam show in below:

$$\text{Adam: } \sum_{\tau=1}^{t} \sigma_{\tau,i} = \sqrt{(1-\beta_2) \sum_{\tau=1}^{\tau=t} \beta_2^{t-\tau} g_{\tau,i}^2} \quad \text{AdaGrad: } \sum_{\tau=1}^{t} \sigma_{\tau,i} = \sqrt{\sum_{\tau=1}^{\tau=t} g_{\tau,i}^2} \tag{14}$$

By guiding with this intuition and our regularization budget definition, we now can conclude that

1. Strictly speaking, non-Adam-Type algorithm such as SGD(M) has only proximal searching region **center at** $x_t$, however, Adam-Type algorithms achieve globally stabilizing regulations via proximal searching region through $\{x_1, x_2, \cdots, x_{t-1}, x_t\}$. Therefore, in our settings, SGD has $\sigma$ function with a constant value where $\{\sigma_\tau = k | \tau = t\}$. A *special case* is when $\eta_t = \frac{1}{t}$ where the SDG has a descending step size $\frac{1}{t}$, we notice that $\sum_{\tau=1}^{t} \sigma_\tau = t$ and $\sigma_t = 1$. In fact, in our theory, we can regard SGD with decreasing learning rate as an adaptive regularization with respect to training iterations.

2. Adam-Type algorithms, in contrast, have stronger bonding constraints with each proximal term $||x - x_\tau||_2^2$. With the training iterations increasing, in order to retain the similar regularization strength, the natural way is increasing the regularization budget $\sum_{\tau=1}^{t} \sigma_\tau$ such that $B_t \geq B_{t-1}$. We have the following comments for Adam:

2.1 Pro: Non-decreasing regularization budget $B_t$ can benefit for Adam, however, unbounded budget causes the infinitely small proximal searching region. On the surface, training will stop without parameters updates. Adam overcomes this problem wisely, exponential moving average (EMA) performs as a sliding window. In other words, the regularization budget is bounded.

2.2 Con: EMA, on the one hand, controls the regularization budget in a range, but it fails to satisfy the primary requirement that regularization budget $B_t$ should be a non-decreasing manner. Lots of previous works point out this issue such as (Reddi et al., 2019; Luo et al., 2019; Chen et al., 2018). However, the way they identify such problems is not natural, for example, the objective function with periodicity gradient rarely seen in real scenarios. Our way seems more easy to access.

| 17 | 122 | 86 | 17 | 134 | 96 | 77 | 31 | 213 |

Figure 1: The number indicates the total swapping by AMSGrad at $v_{t,i}$. The darker heat map colors reveal the lower average swapping interval.

**The non-convergence of Adam** This issue was first identified by Reddi et al. (2019), which points out that the key issue in the convergence of Adam lies in the quantity

$$\Gamma_t = (\frac{\sqrt{v_t}}{\eta_t} - \frac{\sqrt{v_{t-1}}}{\eta_{t-1}}) \tag{15}$$

which assumes to be a non-negative value, but in training, this assumption does not always hold in Adam. Reddi et al. (2019) construct an objective function with periodicity gradient to illustrate the non-convergence of Adam which is hard to follow. And Luo et al. (2019) using a similar way but conduct a heuristic experiment shows that Adam will generate extreme learning rates (also extreme $v_t$ value) that can affect the convergence. Fundamentally, they are identifying the same problem in different expressions. In Reddi et al. (2019) non-convergence Theorem.1 the repeated occurrence gradient $-C$ is, in fact, the extreme learning rate. In our analysis, the emergence of extreme learning rates is due to decreasing regularization budget $B_t - B_{t-1} < 0$ which is equivalent to $\Gamma < 0$, the negative regularization coefficient $\sigma_\tau < 0$ makes the corresponding proximal searching region $||x - x_\tau||_2^2$ to be arbitrarily large when minimizing the Equation. 13.Adam. Again, the parameters behavior in out of the control manner for example $||x - x_\tau||_2^2 \to +\infty$. To this end, Reddi et al. (2019) proposes a very intuitive modification on Adam where

$$v_t = \max\{v_t, v_{t-1}\} \tag{16}$$

Although this setting solves the decreasing regularization budget issue, it still remains the problem so called nonalignment projection, we state this problem in the next section.

### 3.3 NON-ALIGNMENT PROJECTION AND ADAMAL

AMSGrad (Reddi et al., 2019) uses Equation. 16 to ensure the $\Gamma_t \geq 0$ for all $t \in T$. They derive it mainly from an unrealistic objective function which has extreme gradient appearance in periodical manner. Does AMSGrad really outperform Adam or does AMSGrad's design follow the general sketch of Adam-Type algorithm? We conduct an experiment to illustrate the fundamental problem of AMSGrad (or its variant AdaShift). We call this problem as **non-alignment projection**. To illustrate the non-alignment projection problem, we trace a series of entries generated by AMSGrad and sample them from vector $v_{t,i:j}$, our goal is counting the total number of times of that entry being swap with $v_{t,i:j}$ and meanwhile, we record the interval between two swaps. We employ a heat map to visualize this result. As shown in Figure 1, we can find that (1) the different entry has very different swapping counts; (2) the swapping intervals are nonuniform. Note here sampling $v_{t,i}$ from different neuron network layers and different batch size have different results, but we demonstrate the presence of such problems. To be more specific, we present a simple one-step AMSGrad swapping at iteration $t$ and figure out the ill-condition problem.

Suppose in iteration $t$, we receive a corresponding $v_{t,i}$ at $i^{th}$ element. Then in the next step, we would like to perform the update of $v_{t,i}$ to $v_{t+1,i}$ via $v_{t+1,i} = \beta_2 v_{t,i} + (1 - \beta_2)g_{t+1,i}^2$ based on the original design of the Adam. However, if we choose AMSGrad to evaluate $v_{t+2,i}$, we first make the comparison between new $v_{t+1,i}$ and old $v_{t,i}$ to ensure $\Gamma_{t+1}$ to be non-negative and choose $\hat{v}_{t+1,i} = \max\{v_{t+1,i}, v_{t,i}\}$. If this swap happens, we will have $\hat{v}_{t+1,i} = v_{t,i}$ and AMSGrad uses it for $t + 2$ update, and results in

$$v_{t+2,i}^{ams} = \beta_2 \hat{v}_{t+1,i} + (1 - \beta_2)g_{t+2,i}^2 = \beta_2 v_{t,i} + (1 - \beta_2)g_{t+2,i}^2 \tag{17}$$

However, the true step for updating $v_{t+2,i}$ should be

$$v_{t+2,i} = \beta_2 v_{t+1,i} + (1 - \beta_2)g_{t+2,i} = \beta_2^2 v_{t,i} + (1 - \beta_2)\beta_2 g_{t+1,i}^2 + (1 - \beta_2)g_{t+2,i}^2 \tag{18}$$

The difference is quite obvious show in Equation 17 and Equation 18. But another question is Why AMSGrad still working? AMSGrad can still work is because we choose very big $\beta_2$ value as 0.999 which makes the $v_{t+2,i}^{ams} \approx v_{t+2,i}$. However, iterative method will accumulate this small error into

each step and as a consequence, the non-alignment of $v_t$ will lead the model to find a suspicious local optimal. Another non-alignment refers to $m_t$ and $v_k$, AMSGrad updates its $i^{th}$ parameter by

$$x_{t+1,i} = x_{t,i} - \frac{m_{t,i}}{\sqrt{v_{k,i}}} \quad \text{where } k \neq t$$

Recall in the Zinkevich's greedy projection, we minimize the objective loss function relies on current gradient approximation $g_t$ then we perform the projection by projector $\mathcal{P}_{\mathcal{X}}^{-\sqrt{v_k}}$ to resolve the $x_{t+1}$. We see the $v_k$ is not the actual projection matrix for $x_t$ approximate by $m_t$.

To address above issue, recall the define of non-decreasing regularization budget, in Adam setting, $B_t \geq B_{t-1}$ equivalent to $v_t \geq v_{t-1}$ that is

$$v_t - v_{t-1} = (1 - \beta_2)(g_t^2 - v_{t-1}) \geq 0 \Rightarrow g_t^2 - v_{t-1} \geq 0$$

Now, the solution for resolving Adam's non-convergence and non-alignment of AMSGrad is clear. That is before we evaluate $v_t$, we modify $g_t$ to guarantee the non-decreasing condition of $v_{t-1}$ to $v_t$. We illustrate the update detail of AdamAL in Algorithm.1. Using a similar one-step example to

---

**Algorithm 1** AdamAL

1: **Input** $x \in \mathcal{F}$, initial step size $\alpha$, $\beta_1, \beta_2$,
2: set $m_t = 0$, $v_t = 0$
3: **for** t = 1 **to T do**
4:     $g_t = \nabla f_t(x_t)$
5:     $m_t = \beta_1 m_{t-1} + (1 - \beta_1)g_t$
6:     $\hat{g}_t = \max\{g_t^2, v_{t-1}\}$
7:     $v_t = \beta_1 v_{t-1} + (1 - \beta_1)\hat{g}_t$
8:     $x_{t+1} = \mathcal{P}_{\mathcal{X}}^{-\sqrt{v_t}}(x_t - \frac{\eta_t}{\sqrt{v_t}} \odot m_t)$
9: **end for**

---

illustrate how AdamAL can mitigate non-alignment issue when update $v_{t+2}$:

$$v_{t+2}^{adamal} = \beta_2 v_{t+1} + (1 - \beta_2)g_{t+2} = \beta_2^2 v_t + (1 - \beta_2)\beta_2 \hat{g}_{t+1}^2 + (1 - \beta_2)g_{t+2}^2 \tag{19}$$

which reconstruct the $v_{t+2}$ in correct expression. The major difference of AdamAL and AMSGrad is we **do not skip** the gradient update for $v_t$ so that we guarantee the alignment of $g_t$ and $v_t$ so as $m_t$. We have the following bound for AdamAL.

**Theorem 3.1.** *Let $\{x_t\}$ and $\{v_t\}$ be the sequences obtained from AdamAl (Algorithm 1). $\eta_t = \eta/t^{1/2}$, $\beta_{1,t} = \beta_1$, $\beta_{1,t}$ non-decreasing for all $t \in [T]$. Assume that feasible convex set $\mathcal{F}$ such for all $x_t \in \mathcal{F}$ with bounded diameter $D_2$ and $||\nabla f_t(x_t)|| \leq G$. For $x_t$ generated using the AdamAL, we have the following bound on the regret:*

$$\sum_{t=1}^{T} f_t(x_t) - f_t(x^*) \leq \frac{D_2^2}{2\eta_1(1 - \beta_1)} \sum_{i=1}^{d} v_{1,i}^{1/2} + \frac{D_2^2}{2(1 - \beta_1)} \sum_{t=1}^{T} \sum_{i=1}^{d} \frac{v_{t,i}^{1/2}}{\eta_t}$$

$$+ \frac{D_2^2}{2(1 - \beta_1)} \sum_{t=2}^{T} \sum_{i=1}^{d} \{\frac{v_{t,i}^{1/2}}{\eta_t} - \frac{v_{t-1,i}^{1/2}}{\eta_{t-1}}\} \tag{20}$$

$$+ \frac{\eta\sqrt{1 + \log T}}{(1 + \beta_1)(1 - \gamma)\sqrt{(1 - \beta_2)}} \sum_{i=1}^{d} ||g_{1:T,i}||_2$$

## 4 EXPERIMENTS

In this section, we turn to an empirical study of different models to compare new variants with popular optimization methods including SGD(M), Adam, and AMSGrad. We focus on image classification tasks on CIFAR10 and CIFAR100 with different state-of-the-art deep models including ResNet18, ResNet50, VGG16 and VGG19. From the experiment results shown in Figure. 2, we notice that AdamAL outperforms Adam and AMSGrad! This result is desirable because we fix the

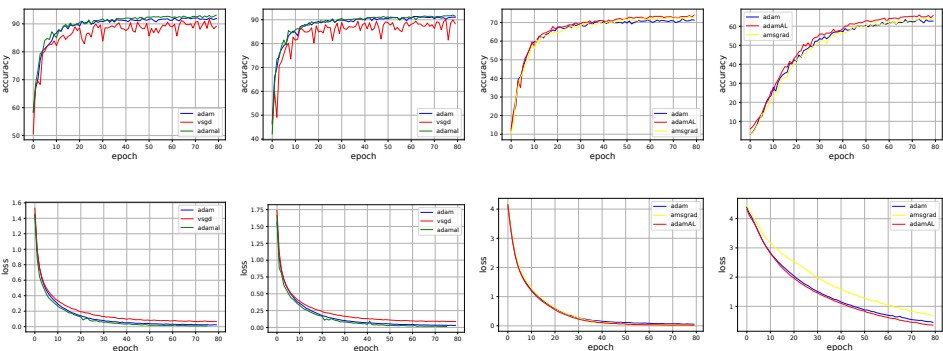

Figure 2: Top: Accuracy of Cifar10 on ResNet18 and VGG16; Cifar100 on ResNet50 and VGG16; Top: Loss of Cifar10 on ResNet18 and VGG16; Cifar100 on ResNet50 and VGG16;(Best see in color)

non-alignment projection issue residing in AMSGrad. In general, from the experiments, AdamAL constantly achieve 1% more accuracy gain than Adam. Notice that we only conduct our experiments with 80 epochs, this is due to that fact that we observe there is no further accuracy improvement without performing any hyperparameter tuning. If we perform hyperparameter tuning, the results show in Table.1. AdamAL can finally reach to 95% accuracy on average on test data, however, the best run of Adam is still lower than AdamAL. It is also worth mentioning that AMSGrad have even worse performance than Adam due to non-alignment. We also compare the result of AdamAL using different mini-batch settings, the result shows that AdamAL is also not sensitive to min-batch size.

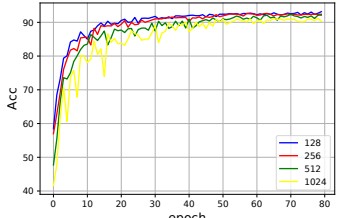

Figure 3: AdamAL in different min-batch

| Algorithm | lr decay | Acc. |
|---|---|---|
| Adam | 75, 125, 175 | 94.98 |
| AMSGrad | 75, 125, 175 | 94.60 |
| AdamAL | 75, 125, 175 | 95.13 |
| VSGD | 75, 125, 175 | 94.73 |

Table 3: hyperparameter tuning, learning rate halve at iteration 75, 125, 175.

## 5 DISCUSS AND CONCLUSION

**The Newton second-order method and** $v_t$ The mystery of $v_t$ in Adam is fascinating. As we discussed in the section 2.2, using second moment to describe $v_t$ runs counter to its primary purpose. Because AdaMax (Kingma & Ba, 2014), p-NosAdam (Huang et al., 2018) and Padam (Zhou et al., 2018a), they apply the p-norm to $v_t$ as $v_t^{1/p}$ and Adam still performs well. Another evidence from AdaShift (Zhou et al., 2018b), in their settings, $v_t$ is no longer acting as second moment, instead they treat it as a learning rate scalar. In our point of view, recall the $\alpha$**-exp-concavity** (Hazan et al., 2007), it structures the relation from first-order to second-order Hessian with $\alpha \leq H/G^2$ where gradient upper bound is $G$ and $H > 0$ is the lower bound of Hessian. This relaxed condition results of $g_t^2$ *to be an possible approximation of Hessian* with scaling factor $\alpha$ at iteration $t$.

**Beyond the Adam-Type method** Assume the above assumption on $g_t^2$ is true, the fast convergence speed of Adam-Type algorithms seems easy to be explained. We also notice that most of the previous works improve the Adam by modifying (or correcting) the term $C$ in Equation 11. Therefore, such corrections cannot speed up the Adam, instead, they define different regularization budget. To make one step furthers to Adam may rely on close approximations of Hessian.

In this paper, we present a new angle to look at the first-order method with adaptive learning rate. We decouple the Adam updating rule as an addition of two regularized terms. In this way, we can identify the intuition behind Adam-Type algorithm. Additionally, we naturally figure out the non-convergence issue resides in AMSGrad and Adam, we propose another variant of Adam algorithm to mitigate such problem.

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

## A  AUXILIARY LEMMAS

**Lemma A.1.** *(McMahan & Streeter, 2010) Let $Q \in S_{++}^n$ with $A = Q^{\frac{1}{2}}$. Let $\mathcal{F}$ be a convex set, and let $u_1, u_2 \in R^n$ with $x_1 = P_{\mathcal{F}}^A(u_1)$ and $x_2 = P_{\mathcal{F}}^A(u_2)$, then*

$$||A(x_2 - x_1)|| \le ||A(u_2 - u_1)|| \tag{21}$$

*Proof.* Define:

$$B(x, u) = \frac{1}{2}||A(x - u)||_2^2 = \frac{1}{2}(x - u)^\top Q(x - u) \tag{22}$$

we can write as:

$$x_1 = \arg\min_{x \in \mathcal{F}} B(x, u_1) \tag{23}$$

Then, note that $\nabla_x B(x, u_1) = Qx - Qu_1$, and **so we must have** $(Qx_1 - Qu_1)^\top(x_2 - x_1) \ge 0$; otherwise for $\delta$ sufficiently small the point $x_1 + \delta(x_2 - x_1)$ would belong to $\mathcal{F}$ (by convexity) and would be closer to $u_1$ than $x_1$. Similarly, **we must have** $(Qx_2 - Qu_2)^\top(x_1 - x_2) \ge 0$. Thus

$$(Qx_1 - Qu_1)^\top(x_2 - x_1) - (Qx_2 - Qu_2)^\top(x_1 - x_2) \ge 0$$
$$-(x_2 - x_1)^\top Q(x_2 - x_1) + (u_2 - u_1)^\top Q(x_2 - x_1) \ge 0$$
$$(u_2 - u_1)^\top Q(x_2 - x_1) \ge (x_2 - x_1)^\top Q(x_2 - x_1)$$

Letting $\hat{u} = u_2 - u_1$, and $\hat{x} = x_2 - x_1$, we have $\hat{x}^\top Q\hat{x} \le \hat{u}^\top Q\hat{x}$. Since $Q$ is semi-definite, we have $(\hat{u} - \hat{x})^\top Q(\hat{u} - \hat{x}) \ge 0$, or equivalently $\hat{u}^\top Q\hat{u} + \hat{x}^\top Q\hat{x} - 2\hat{x}^\top Q\hat{u} \ge 0$. Thus

$$\hat{u}^\top Q\hat{u} \ge -\hat{x}^\top Q\hat{x} + 2\hat{x}^\top Q\hat{u} \ge -\hat{x}^\top Q\hat{x} + 2\hat{x}^\top Q\hat{x} = \hat{x}^\top Q\hat{x} \tag{24}$$

and so

$$||A(u_2 - u_1)||_2^2 = \hat{u}^\top Q\hat{u} \ge \hat{x}^\top Q\hat{x} = ||A(x_2 - x_1)||_2^2 \tag{25}$$

$\square$

## B  PROOF OF THEOREM 3.1

*Proof.* Let $x^* = \arg\min_{x \in \mathcal{F}} \sum_{t=1}^T f_t(x)$, exist and $\mathcal{F}$ is feasible convex set. We have

$$x_{t+1} = P_{\mathcal{F}}^{\sqrt{v_t}}(x_t - \eta_t v_t^{-\frac{1}{2}} \odot m_t) = \min_{x \in \mathcal{F}} ||v_t^{1/4} \odot (x - (x_t - \eta_t v_t^{-\frac{1}{2}} \odot m_t))|| \tag{26}$$

Applying Lemma A.1 we have:

$$
\begin{aligned}
||v_t^{1/4} \odot (x_{t+1} - x^*)||^2 &\le ||v_t^{1/4} \odot (x_t - \eta_t v_t^{-\frac{1}{2}} \odot m_t - x^*)||^2 \\
&= ||v_t^{1/4} \odot (x_t - x^*)||^2 + \eta_t^2 ||v_t^{-\frac{1}{4}} \odot m_t||^2 - 2\eta_t \langle m_t, x_t - x^* \rangle \\
&= ||v_t^{1/4} \odot (x_t - x^*)||^2 + \eta_t^2 ||v_t^{-\frac{1}{4}} \odot m_t||^2 \\
&\quad - 2\eta_t \langle \beta_1 m_{t-1} + (1 - \beta_1)g_t, x_t - x^* \rangle
\end{aligned} \tag{27}
$$

That is

$$
\begin{aligned}
\langle g_t, x_t - x^* \rangle &\le \frac{1}{2\eta_t(1 - \beta_1)}[||v_t^{1/4} \odot (x_{t+1} - x^*)||^2 - ||v_t^{1/4} \odot (x_t - x^*)||^2] \\
&\quad + \frac{\eta_t}{2(1 - \beta_1)}||v_t^{-1/4} \odot m_t||^2 + \frac{\beta_1}{1 - \beta_1}\langle m_{t-1}, x_t - x^* \rangle \\
&\le \frac{1}{2\eta_t(1 - \beta_1)}[||v_t^{1/4} \odot (x_{t+1} - x^*)||^2 - ||v_t^{1/4} \odot (x_t - x^*)||^2] \\
&\quad + \frac{\eta_t}{2(1 - \beta_1)}||v_t^{-1/4} \odot m_t||^2 + \frac{\beta_1}{2(1 - \beta_1)}\eta_t ||v_t^{-1/4} \odot m_{t-1}||^2 + \frac{\beta_1}{2\eta_t(1 - \beta_1)}||v_t^{1/4} \odot (x_t - x^*)||^2
\end{aligned}
$$

We rearrange the Equation 27 to have first inequality. And the second inequality follows the Cauchy-Schwarz and Yong's inequality. Next, we have

$$
\begin{aligned}
\sum_{t=1}^{T} f_t(x_t) - f_t(x^*) &\leq \sum_{t=1}^{T} \langle g_t, x_t - x^* \rangle \\
&\leq \sum_{t=1}^{T} \Big\{ \frac{1}{2\eta_t(1-\beta_1)} [||v_t^{1/4} \odot (x_{t+1} - x^*)||^2 - ||v_t^{1/4} \odot (x_t - x^*)||^2] + \frac{\eta_t}{2(1-\beta_1)} ||v_t^{-1/4} \odot m_t||^2 \quad (28) \\
&\quad + \frac{\beta_1}{2(1-\beta_1)} \eta_t ||v_t^{-1/4} \odot m_{t-1}||^2 + \frac{\beta_1}{2\eta_t(1-\beta_1)} ||v_t^{1/4} \odot (x_t - x^*)||^2 \Big\}
\end{aligned}
$$

The first inequality is due to convexity of function $f$. To bound the above regret, we need the following intermediate result. $\qquad\square$

**Lemma B.1.** *For the parameter settings and conditions assumed in Theorem 3.1, we have:*

$$
\sum_{t=1}^{T} \eta_t ||v_t^{-1/4} \odot m_t||^2 \leq \frac{\eta(1-\beta_1)\sqrt{1+\log T}}{(1+\beta_1)(1-\gamma)\sqrt{(1-\beta_2)}} \sum_{i=1}^{d} ||g_{1:T,i}||_2 \qquad (29)
$$

*Proof.* To prove, we need

$$
\begin{aligned}
\sum_{t=1}^{T} \eta_t ||v_t^{-1/4} \odot m_t||^2 &= \sum_{t=1}^{T-1} \eta_t ||v_t^{-1/4} \odot m_t||^2 + \eta_T \sum_{i=1}^{d} \frac{m_{T,i}^2}{v_{T,i}^{1/2}} \\
&= \sum_{t=1}^{T-1} \eta_t ||v_t^{-1/4} \odot m_t||^2 + \eta \sum_{i=1}^{d} \frac{((1-\beta_1)\sum_{j=1}^{T} \beta_1^{T-j} g_{j,i})^2}{\sqrt{T(\beta_2 v_{t-1,i} + (1-\beta_2)\hat{g}_{t,i})}} \\
&= \sum_{t=1}^{T-1} \eta_t ||v_t^{-1/4} \odot m_t||^2 + \eta(1-\beta_1)^2 \sum_{i=1}^{d} \frac{(\sum_{j=1}^{T} \beta_1^{T-j} g_{j,i})^2}{\sqrt{T(\beta_2 v_{t-1,i} + (1-\beta_2)\hat{g}_{t,i})}} \\
&\leq \sum_{t=1}^{T-1} \eta_t ||v_t^{-1/4} \odot m_t||^2 + \eta(1-\beta_1)^2 \sum_{i=1}^{d} \frac{\sum_{j=1}^{T} \beta_1^{2(T-j)} \sum_{j=1}^{T} g_{j,i}^2}{\sqrt{T(\beta_2 v_{t-1,i} + (1-\beta_2)\hat{g}_{t,i})}} \\
&= \sum_{t=1}^{T-1} \eta_t ||v_t^{-1/4} \odot m_t||^2 + \frac{\eta(1-\beta_1)(1-\beta_1^{2T})}{1+\beta_1} \sum_{i=1}^{d} \frac{\sum_{j=1}^{T} g_{j,i}^2}{\sqrt{T(\beta_2 v_{t-1,i} + (1-\beta_2)\hat{g}_{t,i})}} \\
&= \sum_{t=1}^{T-1} \eta_t ||v_t^{-1/4} \odot m_t||^2 + \frac{\eta(1-\beta_1)(1-\beta_1^{2T})}{1+\beta_1} \sum_{i=1}^{d} \frac{\sum_{j=1}^{T} g_{j,i}^2}{\sqrt{T(1-\beta_2)\sum_{j=1}^{T} \beta_2^{T-j} \hat{g}_{j,i}}} \\
&= \sum_{t=1}^{T-1} \eta_t ||v_t^{-1/4} \odot m_t||^2 + \frac{\eta(1-\beta_1)(1-\beta_1^{2T})}{(1+\beta_1)\sqrt{T(1-\beta_2)}} \sum_{i=1}^{d} \frac{\sum_{j=1}^{T} g_{j,i}^2}{\sqrt{\sum_{j=1}^{T} \beta_2^{T-j} \hat{g}_{j,i}}} \\
&\leq \sum_{t=1}^{T-1} \eta_t ||v_t^{-1/4} \odot m_t||^2 + \frac{\eta(1-\beta_1)(1-\beta_1^{2T})}{(1+\beta_1)\sqrt{T(1-\beta_2)}} \sum_{i=1}^{d} \sum_{j=1}^{T} \frac{g_{j,i}^2}{\sqrt{\beta_2^{T-j} \hat{g}_{j,i}}} \\
&= \sum_{t=1}^{T-1} \eta_t ||v_t^{-1/4} \odot m_t||^2 + \frac{\eta(1-\beta_1)(1-\beta_1^{2T})}{(1+\beta_1)\sqrt{T(1-\beta_2)}} \sum_{i=1}^{d} \sum_{j=1}^{T} \gamma^{T-j} \frac{g_{j,i}^2}{\sqrt{\hat{g}_{j,i}}} \\
&\leq \sum_{t=1}^{T-1} \eta_t ||v_t^{-1/4} \odot m_t||^2 + \frac{\eta(1-\beta_1)(1-\beta_1^{2T})}{(1+\beta_1)\sqrt{T(1-\beta_2)}} \sum_{i=1}^{d} \sum_{j=1}^{T} \gamma^{T-j} |g_{j,i}|
\end{aligned}
$$

The first inequality follows from Cauchy-Schwarz inequality. We let $\gamma = 1/\sqrt{\beta_2}$. The last inequality due to the Algorithm 1 that $\hat{g}_{j,i} = \max\{g_{j,i}, v_{j-1,i}\}$. By using similar upper bounds for all time

steps, the quantity in above can further be bounded as follows:

$$
\begin{aligned}
\sum_{t=1}^{T} ||v_t^{-1/4} \odot m_t||^2 &= \sum_{t=1}^{T} \frac{\eta(1-\beta_1)(1-\beta_1^{2T})}{(1+\beta_1)\sqrt{t(1-\beta_2)}} \sum_{i=1}^{d} \sum_{j=1}^{t} \gamma^{t-j} |g_{j,i}| \\
&= \frac{\eta(1-\beta_1)}{(1+\beta_1)\sqrt{(1-\beta_2)}} \sum_{t=1}^{T} \frac{1-\beta_1^{2t}}{\sqrt{t}} \sum_{i=1}^{d} \sum_{j=1}^{t} \gamma^{t-j} |g_{j,i}| \\
&= \frac{\eta(1-\beta_1)}{(1+\beta_1)\sqrt{(1-\beta_2)}} \sum_{i=1}^{d} \sum_{t=1}^{T} |g_{t,i}| \sum_{j=1}^{t} \frac{(1-\beta_1^{2j})\gamma^{j-t}}{\sqrt{j}} \\
&= \frac{\eta(1-\beta_1)}{(1+\beta_1)\sqrt{(1-\beta_2)}} \sum_{i=1}^{d} \sum_{t=1}^{T} |g_{t,i}| \sum_{j=1}^{t} \frac{(1-\beta_1^{2j})\gamma^{j-t}}{\sqrt{t}} \\
&\leq \frac{\eta(1-\beta_1)}{(1+\beta_1)(1-\gamma)\sqrt{(1-\beta_2)}} \sum_{i=1}^{d} \sum_{t=1}^{T} |g_{t,i}| \sum_{j=1}^{t} \frac{1}{\sqrt{t}} \\
&\leq \frac{\eta(1-\beta_1)}{(1+\beta_1)(1-\gamma)\sqrt{(1-\beta_2)}} \sum_{i=1}^{d} ||g_{1:T,i}||_2 \sqrt{\sum_{j=1}^{t} \frac{1}{t}} \\
&\leq \frac{\eta(1-\beta_1)\sqrt{1+\log T}}{(1+\beta_1)(1-\gamma)\sqrt{(1-\beta_2)}} \sum_{i=1}^{d} ||g_{1:T,i}||_2
\end{aligned}
$$

The first inequality due to the fact that $1 - \beta_1^{2j} < 1$ and $\sum_{j=1}^{T} \gamma^{j-t} \leq 1/(1-\gamma)$. The second inequality is due to simple application of Cauchy-Schwarz inequality. The final inequality is due to the following harmonic sum bound with $\sum_{t=1}^{T} 1/t \leq (1 + \log T)$. $\qquad\square$

Now we return to the proof of Theorem 3.1, by using above lemma, we have:

$$
\begin{aligned}
\sum_{t=1}^{T} f_t(x_t) - f_t(x^*) &\leq \sum_{t=1}^{T} \langle g_t, x_t - x^* \rangle \\
&\leq \sum_{t=1}^{T} \{ \frac{1}{2\eta_t(1-\beta_1)} [||v_t^{1/4} \odot (x_{t+1} - x^*)||^2 - ||v_t^{1/4} \odot (x_t - x^*)||^2] \\
&\quad + \frac{\beta_1}{2\eta_t(1-\beta_1)} ||v_t^{1/4} \odot (x_t - x^*)||^2 \} + \frac{\eta\sqrt{1+\log T}}{(1+\beta_1)(1-\gamma)\sqrt{(1-\beta_2)}} \sum_{i=1}^{d} ||g_{1:T,i}||_2 \\
&= \frac{1}{2\eta_1(1-\beta_1)} ||v_t^{1/4} \odot (x_1 - x^*)||^2 + \frac{1}{1-\beta_1} \sum_{t=1}^{T} \frac{1}{2\eta_t} ||v_t^{1/4} \odot (x_t - x^*)||^2 \\
&\quad + \frac{1}{2(1-\beta_1)} \sum_{t=2}^{T} \{ \frac{||v_t^{1/4} \odot (x_t - x^*)||^2}{\eta_t} - \frac{||v_{t-1}^{1/4} \odot (x_t - x^*)||^2}{\eta_{t-1}} \} \\
&\quad + \frac{\eta\sqrt{1+\log T}}{(1+\beta_1)(1-\gamma)\sqrt{(1-\beta_2)}} \sum_{i=1}^{d} ||g_{1:T,i}||_2 \\
&= \frac{1}{2\eta_1(1-\beta_1)} \sum_{i=1}^{d} v_{1,i}^{1/2}(x_{1,i} - x_{\cdot,i}^*)^2 + \frac{1}{2(1-\beta_1)} \sum_{t=1}^{T} \sum_{i=1}^{d} \frac{v_{t,i}^{1/2}(x_{t,i} - x_{\cdot,i}^*)^2}{\eta_t} \\
&\quad + \frac{1}{2(1-\beta_1)} \sum_{t=2}^{T} \sum_{i=1}^{d} (x_{t,i} - x_{\cdot,i}^*)^2 \{ \frac{v_{t,i}^{1/2}}{\eta_t} - \frac{||v_{t-1,i}^{1/2}}{\eta_{t-1}} \} \\
&\quad + \frac{\eta\sqrt{1+\log T}}{(1+\beta_1)(1-\gamma)\sqrt{(1-\beta_2)}} \sum_{i=1}^{d} ||g_{1:T,i}||_2
\end{aligned}
\tag{30}
$$

For the sake of clarity, we further suppose that the feasible region of parameter $x$ with bound $D_2$, the Equation 30 becomes to:

$$\sum_{t=1}^{T} f_t(x_t) - f_t(x^*) \leq \frac{D_2^2}{2\eta_1(1-\beta_1)} \sum_{i=1}^{d} v_{1,i}^{1/2} + \frac{D_2^2}{2(1-\beta_1)} \sum_{t=1}^{T}\sum_{i=1}^{d} \frac{v_{t,i}^{1/2}}{\eta_t}$$

$$+ \frac{D_2^2}{2(1-\beta_1)} \sum_{t=2}^{T}\sum_{i=1}^{d} \{\frac{v_{t,i}^{1/2}}{\eta_t} - \frac{||v_{t-1,i}^{1/2}}{\eta_{t-1}}\} \tag{31}$$

$$+ \frac{\eta\sqrt{1+\log T}}{(1+\beta_1)(1-\gamma)\sqrt{(1-\beta_2)}} \sum_{i=1}^{d} ||g_{1:T,i}||_2$$

Therefore, we finish the proof of Theorem 3.1 as desired.

