# OpenReview forum: "A new perspective in understanding of Adam-Type algorithms and beyond"
_ICLR.cc/2020/Conference — Reject_

### Official Review · AnonReviewer3 · 2019-10-22
**Official Blind Review #3**

**Rating:** 3

**Review:**

Summary:

This work proposed a framework to analyze both Adam-type algorithms and SGD-type algorithms. The authors considered both of them as specialized cases of mirror descent algorithms and provided a new algorithm AdamAL. The authors showed experiments to backup their theoretical results.

Pros:

The authors provided a novel framework to analyze Adam-type algorithms by using standard FTRL framework. It provides a unified viewpoint to consider a broad class of algorithms. The authors also provided a new algorithm AdamAL to overcome some shortcomings in previous algorithms.

Cons:

- The novelty of this paper is limited. The authors provided a framework to analyze Adam-type algorithms. However, it seems that contribution is more conceptual rather than practical. I suggest the authors add more examples or theorems to show the superiority of their mirror descent framework.
- Some of the explanations in this paper may be wrong. For instance, around equation 17, the authors suggested that if the swap happens, then v_{t+1} = v_t. That is not correct since the swap happens for each coordinate. Meanwhile, the explanation about why AdamAL is better than AMSgrad is quite poor since the update rule of AMSgrad can also guarantee the coordinate decreasing of v_t. I suggest the authors explain more on the algorithm design.
- The authors should complete the proof of Theorem 3.1.
- The settings of experiments are limited. The authors should at least compare AdamAL with other baseline algorithms on some modern deep learning tasks including Imagenet.

Minor comments:

- Below equation 8, detail -> details.
- The authors should add the definition of 1:t in subscript for g_{1:t} or \phi_{1:t}.
- Page 6, the first paragraph, logt-> \log t
- This paper lacks some references in this area.

J. Chen and Q. Gu. Closing the generalization gap of adaptive gradient methods in training
deep neural networks. arXiv preprint arXiv:1806.06763, 2018.
Ward, R., Wu, X. and Bottou, L. (2018). Adagrad stepsizes: Sharp convergence over nonconvex
landscapes, from any initialization. arXiv preprint arXiv:1806.01811 .
Li, X. and Orabona, F. (2018). On the convergence of stochastic gradient descent with adaptive
stepsizes. arXiv preprint arXiv:1805.08114 .

**Experience Assessment:**

I have published one or two papers in this area.

**Review Assessment: Checking Correctness Of Derivations And Theory:**

I carefully checked the derivations and theory.

**Review Assessment: Checking Correctness Of Experiments:**

I carefully checked the experiments.

**Review Assessment: Thoroughness In Paper Reading:**

I read the paper at least twice and used my best judgement in assessing the paper.

---

> ### Author Response · Authors · 2019-11-07
> **RE: Official Blind Review #3**
>
> We thank the reviewer for their thoughtful comments and questions. We address them in order.
>
> 1. The novelty of this paper is limited.
> We restate our contributions in the new version in Section 1. We summarize our contribution as follows:
> We provide a new perspective in understanding the non-convergence behavior of Adam-Type algorithms based on mirror descent approach. Our analysis agrees well with the previous works but much more intuitive and effective.
> By using our analyzing framework, we can:
> 2a) clearly identify the functionality of each part of Adam algorithm such as \beta_1,2, v_t and m_t (Section 3.1)
> 2b) easily explain why EMA can help Adam-type algorithms with smooth trajectory and less oscillation around the optimal point.
> 3c) the sensitivity of loss function and so on.
> Based on our observation, we identify potential faults in Adam-Type algorithms and we provide a new Adam variant algorithm, named AdamAL.
> We conduct a series of experiments on different machine learning tasks and models by using our AdamAL algorithm, and the results are promising and the performance of AdamAL is never worse than Adam.
> In fact, the primary goal of our work is trying to understand the mechanism of Adam-type algorithms, especially the v_t. The mystery of Adam’s v_t draws a lot of attention in recent years, such as Zhou et al.2018, Balles et al. 2018. However, the traditional analysis is insufficient, it regards the v_t as second momentum which is hard to understand. Besides v_t, we also use our analyzing framework to explore the usage of hyper-parameters \beta_1 and \beta_2. It is also a hot topic in the recent (Lucas et al. 2019).
>
> You also mentioned that “It provides a unified viewpoint to consider a broad class of algorithms.”. Yes, we think our main contribution aims to deliver a new perspective in understanding of Adam-type algorithms. Based on our knowledge, we do not know any previous work similar to us. In addition, our new algorithm adamAL is a very good example to show our framework is useful. Based on the principle of our framework, we design AdamAL in a very natural way. Meanwhile, we can use this framework to design more Adam-Type algorithms. This also leads to another contribution of our work, that is, our framework could help researchers to design a new and better algorithm in the future.
>
> “It seems that contribution is more conceptual rather than practical”. We mentioned this in the above. Again, AdamAL is one of the practical algorithms we design by the guidance of our unified framework.
>
> “I suggest the authors add more examples or theorems to show the superiority of their mirror descent framework.” Thank you for your valuable suggestion. In fact, we are trying to show the superiority of our framework in Corollary 1 and 2; and we explain some of them in the Section 3.1 as well. We plan to extend more algorithms and theorems according to this framework in the future.
>
>
> 2. "Some of the explanations in this paper may be wrong. For instance, around equation 17, the authors suggested that if the swap happens, then v_{t+1} = v_t. That is not correct since the swap happens for each coordinate."
>
> We actually mentioned that the \max(x, y) operation denotes the entry-wise maximum in Section 2 Notations which means if swap happens, v_{t+1} = \max{v_{t+1}, v_t} will perform entry-wise swap.
> Thank you for pointing out this. We rewrite the Section 3.3 and we use entry-wise notation such as v_{t+1, i}.
>
> 3. "Meanwhile, the explanation about why AdamAL is better than AMSgrad is quite poor since the update rule of AMSgrad can also guarantee the coordinate decreasing of v_t."
>
>
> 4. The authors should complete the proof of Theorem 3.1.
> We add proof in Appendix.
>
> 5. The settings of experiments are limited.
> We conduct more experiments and show the results in this version, please check.
>
> Minor comments:
>
> 1. Below equation 8, detail -> details.
> Thank you very much, we fix it in new revision.
>
> 2. The authors should add the definition of 1:t in subscript for g_{1:t} or \phi_{1:t}.
> Thank you for your suggestion, we add those definitions in the Section 2 Notations.
>
> 3. Page 6, the first paragraph, logt-> \log t
> Thank you very much, we fix it in new revision.
>
> 4. This paper lacks some references in this area.
> Thank you for your suggestion, we add these references to this revision.
>
>
> References:
> Zhou et al.2018: https://openreview.net/forum?id=HkgTkhRcKQ
> Balles et al. 2018: https://arxiv.org/pdf/1705.07774.pdf
> Lucas et al. 2019: https://openreview.net/forum?id=Syxt5oC5YQ

---

### Official Review · AnonReviewer1 · 2019-10-26
**Official Blind Review #1**

**Rating:** 1

**Review:**

This paper analyzed a few issues of Adam, and proposed a new variant of Adam called AdamAL.

There are quite a few issues:
--It is not clear whether the heuristic observations are useful.
--Theorem 3.1 does not even lead to the convergence of the algorithm.
--The simulation does not show the advantage.
--There are too many typos and grammar errors.

1) Heuristic observations.
One main observation is that v_{t+1} = beta_2 v_t + (1 - beta_2) g_{t+1}^2 in the original Adam will be different from \hat{v}_t defined in AMSGrad. The paper states "...will accumulate this small error into each step" ..."will lead this model to find a suspicious local optimal".  This claim makes little sense. Even if AMSGrad uses a different v from the original Adam, it is not necessarily bad. In addition, why is this related to "suspicious local optimal" (I suppose this paper intends to say "spurious local optima")? I do not have any intuition why this is related to spurious local minima.



2) Too many typos and grammar errors.

Just give one example. In the first paragraph of Sec. 3.3, there are at least 15 typos, and a few sentences are hard to read:
   "They derive it mainly from an unrealistic objective function" --what does "objective function" mean?
   "Does it really solve the problem or dose this design violate the intuition of Adam-Type algorithm"?  --what is the "intuition" to be violated? I roughly get the point of this sentence, but "intuition" is not a good word here.
   "To be more specifically, we present a simple one-step AMSGrad swapping at iteration t and figure out the ill-condition problem". --What is "ill-condition problem"? It is not mentioned in this paragraph, and I don't know where it comes from.
   In the whole paper, there are too many problematic sentences to enumerate (a very rough estimate: at least 30?)  It makes the paper very difficult to read.





**Experience Assessment:**

I have published one or two papers in this area.

**Review Assessment: Checking Correctness Of Derivations And Theory:**

I assessed the sensibility of the derivations and theory.

**Review Assessment: Checking Correctness Of Experiments:**

I assessed the sensibility of the experiments.

**Review Assessment: Thoroughness In Paper Reading:**

I read the paper at least twice and used my best judgement in assessing the paper.

---

> ### Author Response · Authors · 2019-11-07
> **RE:Official Blind Review #1**
>
> We thank the reviewer for their time and their kind words about our work. We will address your concerns and questions in the order you wrote them.
>
> 1.It is not clear whether the heuristic observations are useful.
> I am assuming that heuristic observations refer to the observations in Section 3.3 and Figure 1 (if not this one, please correct me). The observations in Section 3.3 demonstrate that (1) different coordinates have very different swapping counts; (2) the swapping frequency is nonuniform, which means the swapping interval between two swaps happens is different for different coordinates. They are very useful because they identify the non-alignment issue in AMSGrad.
>
> 2.Theorem 3.1 does not even lead to the convergence of the algorithm.
>
> Theorem 3.1 shows the convergence of a class of Adam-Type algorithms for the non-convex optimization. And we think it leads to the convergence of the algorithm (See https://arxiv.org/abs/1808.02941 for reference). In our first submitted paper, we actually follow the same proof sketch above. We would like to check and fix the problems of our proof, and thank you for pointing out this issue.
>
> However, in this new revision, we decide to replace Theorem 3.1 by using Zinkevich regret analysis which is commonly used in proving the Adam-Type algorithms (Reddi et al. 2019, Chen et al. 2018) and it is sufficient for proving our algorithm.
>
> 3.The simulation does not show the advantage.
>
> We are sorry for the inappropriate representation of our experiment results. We conduct experiments on ResNet18 and VGG16 with CIFAR10, and in fact, our algorithm shows the advantage of such settings, as we mentioned in Section 4, AdamAL's performance is never worse than Adam, AMSGrad and Vanilla SGD. When you zoom in on the last 20 epochs, AdamAL achieves better accuracy on validation set and meanwhile, it achieves lower loss values on training set.
>
> In order to see our experiment results better, in this revision:
> (a) We will zoom in on both training and testing results for clear visualization in next version.
> (b) We add more convincing experimental results to our paper.
>
>
> 4.There are too many typos and grammar errors.
> We fix them in this revision and thank you for pointing out.
>
> 1) Heuristic observations.
> In Section 3.3, we use one-step analysis to illustrate the two different approaches. For Adam the update of v_{t+2, i} at i-th coordinate is v_{t+2, i} = \beta_2^2 v_{t,i} + (1-\beta_2)\beta_2 g_{t+1, i}^2 + (1-\beta_2) g_{t+2, i}^2. However, the AMSGrad uses v_{t+2, i} = \beta_2 v_{t, i} + (1-\beta_2) g_{t+2, i}^2 if swap happens. It is very clear that AMSGrad will not use the information of g_{t+1, i} in only one-step update. And we also notice that Adam use \beta_2^2v_{t,i} and AMSGrad use \beta_2 v_{t, i}. We can not ignore this difference even though we have big \beta_2 value 0.999. Because the experiment in Section 3.3 (Figure 1) tells us the different coordinates have very different swapping counts and there is no way we can know when the swap happens and how many swaps will it be. Suppose v_t = (v_{t, 1}, v_{t, 2}, ... v_{t, i}, ...), and v_{t, i} is the i-th coordinate at iteration t. AMSGrad will generate v_t = (..., v_{t, i}, v_{*, j}, ...) with j-th coordinate using some value v_{*, j} but not v_{t, j} since the the ASMGrad swaps happen in nearly random behavior. This will cause the issue which we call non-alignment of v_t coordinates. The value of v_{*, j} is unpredictable and totally loses its meaning. If we use incorrect (or not up-to-date) coordinates of v_t, the searching direction of some coordinates to the optimal may be wrong, which highly leads the result to the suspicious local optimal. The wrong searching direction due to (1)unpredictable swap; (2)g_{, i} the gradient information skipping.
>
> 2) Too many typos and grammar errors.
>
> 2a) "They derive it mainly from an unrealistic objective function" -- what does "objective function" mean?
>
> The "objective function" refers to a loss function constructed by Reddi et al. (2019) in Section 3 of the Theorem.1. They use this function to prove the non-convergence of Adam. We mention the Reddi et al's objective function in Section 3.2 "the objective function with periodicity gradient rarely seen in real scenario." And in the section "The non-convergence of Adam" with "Reddi et al. (2019) construct an objective function with periodicity gradient to illustrate ... which is hard to follow.". Because this paper is well-known and it is the best paper of ICLR 2018, we prefer the readers to check it from the reference for the sake of simplicity. Again, the Reddi et al's objective function refers to function f_t(x) = Cx for t mod 3 = 1 or f_t(x) = -x otherwise.
>
> 2b)What is the "ill-condition problem"?
> We refer to the AMSGrad non-alignment problem. We remove the "ill-condition problem" in the revision.
>
> References:
> Reddi et al. 2019: https://arxiv.org/pdf/1904.09237.pdf
> Chen et al. 2018: https://arxiv.org/pdf/1902.09843.pdf

---

### Official Review · AnonReviewer2 · 2019-10-28
**Official Blind Review #2**

**Rating:** 3

**Review:**

The paper proposes to study some weaknesses of Adam and AMSGrad and propose a new method called AdamAL that is evaluated on CIFAR10.

I am not from this area, but unfortunately I find this paper to not be rigorously written or organized. Section 3.3 for instance which discusses the non-alignment projection issue with AMSGrad is not rigorously written. There are no proofs to any theorems and even some of the theorems/corollaries are not written rigorously.

Organization-wise I feel it is difficult to see where the paper is going and some sort of outline / notation box would help.

Empirically, there are results on one dataset CIFAR10 that shows the author's proposed variant AdamAL works better. However, it is not the detailed types of experiments I was expecting of a paper that points out specific issues in AMSGrad. Shouldn't the experiments be showing these weaknesses in some sort of controlled setting?


**Experience Assessment:**

I do not know much about this area.

**Review Assessment: Checking Correctness Of Derivations And Theory:**

I assessed the sensibility of the derivations and theory.

**Review Assessment: Checking Correctness Of Experiments:**

I assessed the sensibility of the experiments.

**Review Assessment: Thoroughness In Paper Reading:**

I read the paper at least twice and used my best judgement in assessing the paper.

---

> ### Author Response · Authors · 2019-11-07
> **We appreciate your thorough review and helpful suggestions.**
>
> We thank the reviewer for their time and their kind words about our work.
>
> 1. We upload the revised version to resolve all of your concerns.
>
> 1.1 We proofread the paper and complete the proof of our main Theorem 3.1.
> 1.2 We rewrite the Section 3.3 for more clear heuristic observations on the ASMGrad algorithm. We also carefully explain the differences of AMSGrad and desired Adam algorithms.
> 1.3 We restate the contribution of our work in the introduction section. You will be easy to follow the outline of our paper.
> 1.4 Notations are explained in Section 2 Notations. In this version, we add more notation definitions for your information.
> 1.5 We add more detailed experiment results in this revision, and we also conduct more experiments with different settings and algorithms.
>
> 2. "Empirically, there are results on one dataset ... AdamAL works better. However, it is not the detailed type of experiments I was expecting of a paper that points out specific issues in AMSGrad.
>
> To point out the issues in AMSGrad, we first conduct the experiment on sampled coordinates of v_{t, i:j}. More specific, we randomly sampled 10 coordinates v_{t, k} where i<=k<=j. We trace those coordinates at each iteration. Here, two metrics are being used, one is the total number of swapping and another is the frequency in between the two swaps. We have two obversions: (1) different coordinates have very different swapping counts; (2) the swapping frequency is nonuniform, which means the swapping interval between two swaps happens is different for different coordinates. (Mentioned in Section 3.3)
>
> These two observations may cause the performance of AMSGrad not better than Adam. And AMSGrad is also suffering the issue of poor generalization ability as the Adam. The reasons are: (1) It is necessary to keep all the coordinates of v_t to be up-to-date. However, AMSGrad breaks this rule. The example shown in Section 3.3 tells us that AMSGrad has v_{t+2, i} = \beta_2 v_{t, i} + (1-\beta_2)g_{t+2, i}^2 (Equation 17) if the swap happens. The gradient information of g_{t+1, i} will be skipped. This may cause the slowing of convergence speed or become a possible reason for poor generalization ability. (2) All coordinates in v_t should stay on the same iteration of t, but AMSGrad can cause non-alignment of v_t coordinates, for example, using v_{t, i} and v_{t-m, j} in the same iteration. (3) The swapping frequency is nonuniform and unpredictable. When the training iterations are increasing, the small errors will be accumulated at each iteration and affect the final model accuracy.
>
> Yet, the above issues may not cause the non-convergence issue of AMSGrad, but the truths are (1) AMSGrad does not show superiority than Adam; (2) AMSGrad also suffer the poor generalization ability issue, same as Adam; (3)"Even if AMSGrad uses a different v from the original Adam, it is not necessarily bad."(Review #1), we agree with this point of view, however, in our experiment results, adamAL performance better than both Adam and AMSGrad.
>
> Back to your question: "Shouldn't the experiments be showing these weaknesses in some sort of controlled setting?". We basically discover this weakness by conducting the tracing experiments on sampled coordinates of v_t, and we justify the problem of AMSGrad by Equation 17 and Section 3.3. We will do some research on this issue in the future.
>
> Besides your questions, we would like to restate our main contribution of our work, because it seems all reviews are focusing on our algorithm part and ignoring the way we try to understand the Adam-Type algorithms. In fact, in this work, we try to deliver a new perspective in understanding of Adam-type algorithms which no one did this before. Based on our analysis framework, we can clearly identify the functionality of each part of Adam algorithm such as v_t and m_t. We can also easily explain why EMA can help Adam-type algorithms with smooth trajectory and so on. In contrast, the traditional frameworks have difficulty to demonstrate these. The results produced by using our analyze framework are quite promising and they are consistent with many other research works (mentioned in Section 3.1). Our main contribution is providing a unified viewpoint to Adam-type algorithms which can help researchers to design a better algorithm in the future. Our new algorithm adamAL is a very good example to show our framework is useful. We directly design this algorithm by using our analyze framework and as a result, AdamAL outperforms AMSGrad and Adam.

---

### Decision · Program_Chairs · 2019-12-19

**Decision:**

Reject

**Comment:**

In this paper, the authors draw upon online convex optimization in order to derive a different interpretation of Adam-Type algorithms, allowing them to identify the functionality of each part of Adam. Based on these  observations, the authors derive a new Adam-Type algorithm,  AdamAL and test it in 2 computer vision datasets using 3 CNN architectures. The main concern shared by all reviewers is the lack of novelty but also rigor both on the experimental and theoretical justification provided by the authors. After having read carefully the reviews and main points of the paper, I will side with the reviewers, thus not recommending acceptance of this paper.